# Physiological and Molecular Mechanisms of Rice Tolerance to Salt and Drought Stress: Advances and Future Directions

**DOI:** 10.3390/ijms25179404

**Published:** 2024-08-29

**Authors:** Qingyang Li, Peiwen Zhu, Xinqiao Yu, Junying Xu, Guolan Liu

**Affiliations:** 1College of Agriculture, Yangtze University, Jingzhou 434025, China; lqy13437246041@126.com; 2Shanghai Agrobiological Gene Center, Shanghai 201106, China; sagczpw@gmail.com (P.Z.); yuxq66@126.com (X.Y.)

**Keywords:** salt and drought stress, physiological and molecular mechanisms, transcription factors, gene editing, multi-omics, smart agriculture

## Abstract

Rice, a globally important food crop, faces significant challenges due to salt and drought stress. These abiotic stresses severely impact rice growth and yield, manifesting as reduced plant height, decreased tillering, reduced biomass, and poor leaf development. Recent advances in molecular biology and genomics have uncovered key physiological and molecular mechanisms that rice employs to cope with these stresses, including osmotic regulation, ion balance, antioxidant responses, signal transduction, and gene expression regulation. Transcription factors such as DREB, NAC, and bZIP, as well as plant hormones like ABA and GA, have been identified as crucial regulators. Utilizing CRISPR/Cas9 technology for gene editing holds promise for significantly enhancing rice stress tolerance. Future research should integrate multi-omics approaches and smart agriculture technologies to develop rice varieties with enhanced stress resistance, ensuring food security and sustainable agriculture in the face of global environmental changes.

## 1. Introduction

Salt and drought stress are among the most significant abiotic stresses that threaten global agricultural production, particularly when they occur simultaneously. These environmental challenges are not only widespread but also highly destructive, often leading to severe reductions in crop yield and quality. Drought exacerbates the impact of salt stress, creating a compounded effect that further hinders crop growth and productivity. This synergy between salt and drought stress can be particularly devastating in regions where agriculture is already vulnerable due to harsh climatic conditions [1]. Globally, over 800 million hectares of land are affected by salinity, with the problem being most acute in arid and semi-arid regions. In these areas, the scarcity of water resources exacerbates the issue, leading to the coexistence of salt and drought stresses. This double burden on crops results in significant losses in agricultural output, making it increasingly difficult to sustain food production in affected regions [2,3].

Rice (*Oryza sativa* L.) is one of the world’s most important staple crops, providing nourishment to more than half of the global population. Its role in global food security is critical, especially in densely populated regions where rice is a primary source of calories and nutrition [4]. Despite the substantial increases in rice production since the Green Revolution, current yields are still insufficient to meet the demands of a rapidly growing global population. As the world faces the dual challenges of population growth and climate change, enhancing rice yields and expanding cultivation into marginal areas have become urgent strategies to ensure global food security [5]. Rice exhibits a moderate level of tolerance to salt, and this characteristic has driven significant research into the development and cultivation of salt-tolerant rice varieties. These varieties are particularly valuable in regions with saline-alkaline soils, where conventional rice cultivation is often limited by the harsh growing conditions [6]. However, rice grown in inland saline-alkaline areas frequently encounters the dual challenges of salinity and drought stress, which can severely stunt growth and reduce yields. This dual stress condition highlights the need for a comprehensive understanding of the physiological and molecular mechanisms that underlie rice’s responses to these stresses. Such knowledge is essential for developing new strategies to enhance rice productivity under adverse environmental conditions [7].

In recent years, advances in molecular biology, genetics, and biotechnology have significantly deepened our understanding of how rice copes with salt and drought stress. These scientific breakthroughs have revealed the complex network of genes and molecular pathways that are activated in response to stress, providing new insights into the mechanisms that enable rice to survive and even thrive under challenging conditions. This review aims to summarize the latest research progress on rice responses to salt and drought stress, with a focus on key genes, molecular mechanisms, and their potential applications in breeding practices (Figure 1). The findings discussed not only enhance our understanding of the fundamental processes of stress resistance in rice but also offer valuable theoretical foundations for breeding rice varieties that are both salt-tolerant and drought-resistant, thereby contributing to global food security.

## 2. Impact of Salt and Drought Stress on Rice

### 2.1. Effects on Morphology, Physiology, Biochemistry, and Yield

Salt and drought stress significantly affect the growth and development of rice (Figure 2). Salt stress primarily influences plants through ionic toxicity and osmotic stress, leading to cellular ion imbalance and increased osmotic pressure, which in turn cause oxidative stress and lipid peroxidation in membranes [8]. Throughout the entire growth period of rice, salt stress hinders various stages of development. During the germination stage, salt stress inhibits seed germination [9]. In the seedling stage, rice growth slows, leaves lose their green color, dry weight and fresh weight decrease, and survival rates drop [8]. As rice enters the reproductive stage, salt stress more severely inhibits both the aerial and subterranean parts of the plant, resulting in reduced plant height, shorter root length, limited tillering, yellowing and wilting of leaves, and suppressed photosynthesis [10]. Ultimately, these impacts lead to spikelet sterility, reduced fertility rates, fewer effective panicles, and lower thousand-grain weight, severely affecting yield [11].

Physiologically and biochemically, salt stress triggers a series of complex responses, including ion imbalance, reactive oxygen species (ROS) accumulation, increased osmolyte levels, and hormonal changes [8]. Salt stress raises the concentration of Na^+^ in cells, disrupting the normal function of K^+^ and affecting cellular metabolism [12]. It induces excessive ROS production, leading to oxidative stress and lipid peroxidation, damaging cell membranes, proteins, and DNA [13]. To cope with osmotic stress, rice accumulates significant amounts of proline, betaine, and soluble sugars, helping cells maintain osmotic balance [14]. Simultaneously, salt stress significantly elevates abscisic acid (ABA) levels [15], regulating stomatal closure to reduce water transpiration and promoting root growth [16,17]. Additionally, rice increases the levels of antioxidant enzymes (such as superoxide dismutase, peroxidase, and catalase) and non-enzymatic antioxidants (such as ascorbic acid and glutathione) to remove excess ROS and mitigate oxidative damage [18].

Drought stress primarily affects the water status and balance within plants, significantly impacting both the aerial and subterranean parts of rice [8]. Under drought conditions, intracellular water content decreases, turgor pressure drops, causing wilting and slow growth. Under mild drought stress, rice adjusts the water potential inside and outside cells through osmotic regulation and increases cell wall elasticity to maintain normal root growth [19]. However, under moderate to severe drought stress, lateral root growth is significantly inhibited, total root length decreases, and root development is severely affected, further exacerbating the negative impact of water deficiency on overall growth [20]. Drought stress also noticeably affects the aerial parts of rice, significantly reducing plant height, leaf area, tiller number, and biomass [21]. Leaves curl and yellow; photosynthetic capacity weakens; young panicle differentiation is hindered; and fertility rates, effective panicle numbers, and thousand-grain weight decrease; ultimately leading to a significant decline in yield [22]. Stomatal closure reduces stomatal conductance, thereby reducing transpiration to conserve water, but also limiting CO_2_ absorption, further inhibiting photosynthesis [23].

On the physiological and biochemical level, drought stress induces a series of complex reactions. Firstly, stomatal conductance decreases, reducing water transpiration and CO_2_ absorption [24]. Secondly, drought conditions lead to ROS accumulation, causing oxidative stress and damaging cell membranes, proteins, and DNA [25]. To cope with drought stress, the levels of osmolytes such as proline, betaine, and soluble sugars in rice cells increase to maintain osmotic balance [26]. Additionally, drought stress significantly elevates ABA levels, regulating stomatal closure and promoting root growth [27]. The levels of antioxidant enzymes (such as superoxide dismutase, peroxidase, and catalase) and non-enzymatic antioxidants (such as ascorbic acid and glutathione) also increase to remove excess ROS and mitigate oxidative damage [28].

Both salt and drought stress significantly affect rice growth and development, primarily through impacts on water status, ion balance, and physiological and biochemical processes. Morphologically, both stresses lead to significant reductions in plant height, leaf area, tiller number, and biomass, with leaves wilting and yellowing. Physiologically and biochemically, both stresses reduce stomatal conductance, limiting water transpiration and CO_2_ absorption, affecting photosynthesis.

Additionally, both stresses lead to ROS accumulation, inducing oxidative stress and damaging cell membranes, proteins, and DNA. To mitigate these stresses, rice accumulates osmolytes (such as proline, betaine, and soluble sugars) and increases the levels of antioxidant enzymes and non-enzymatic antioxidants, alleviating damage. Ultimately, these stresses result in reduced yield and quality of rice.

### 2.2. Impact on Rice Quality

When discussing the morphological, physiological, biochemical, and yield changes in rice under salt and drought stress, it is equally important to consider the changes in rice quality. Rice quality is influenced by multiple factors, including variety, climate change, irrigation conditions, and cultivation practices. These factors become more difficult to control under stress conditions, resulting in limited research on rice quality under stress conditions.

We particularly focus on the changes in the nutritional composition of rice under salt stress. Saline-alkali soils are rich not only in Na^+^ and Cl^−^ but also in Ca^2+^, Mg^2+^, and various trace elements such as Fe^2+^, Mn^2+^, Zn^2+^, and Cu^2+^. Does rice grown in saline-alkali soils accumulate more trace elements in the grains? Does such rice have higher nutritional value? These questions are worthy of in-depth research and discussion. Numerous studies have shown that rice under salt stress exhibits quality changes characterized by low starch, high protein, and multiple nutrient elements [29,30]. These changes provide new ideas for the development and utilization of saline-alkali land, such as breeding functional rice with high protein and multi-mineral nutrition as selling points, which is beneficial for market promotion and further advances in related research.

Research on drought stress is mainly conducted under normal conditions, focusing on indicators such as chalkiness and brown rice rate. There is limited research on changes in the nutritional composition of rice under drought stress and even fewer studies on rice under combined salt and drought stress. Therefore, research in this area has significant scientific and practical value.

## 3. Mechanisms of Rice Responses to Salt and Drought Stress

Rice exhibits a series of typical response mechanisms under salt stress. Firstly, it eliminates or isolates toxic ions through ion balance systems such as the sodium-potassium pump, salt glands, and sodium transport proteins [31]. Secondly, it maintains cellular osmotic pressure by accumulating osmolytes such as proline and betaine [32]. Additionally, rice increases the levels of antioxidant enzymes such as superoxide dismutase (SOD), peroxidase (POD), and catalase (CAT), as well as non-enzymatic antioxidants like ascorbic acid and glutathione, to remove reactive oxygen species (ROS) and protect cells from oxidative damage [33]. Rice also enhances stress tolerance by activating transcription factors (TFs) to regulate stress-responsive genes and through hormone regulation of growth and development. Under drought stress, rice exhibits similar osmotic regulation, antioxidant regulation, transcriptional regulation, and hormone regulation mechanisms as seen under salt stress, along with unique stomatal regulation (Figure 3) [34].

### 3.1. Synthesis and Accumulation of Osmolytes

To mitigate osmotic stress, rice synthesizes and accumulates osmolytes such as proline, betaine, and soluble sugars in response to both salt and drought conditions. These osmolytes play a crucial role in maintaining cellular osmotic balance and protecting cellular structures from the detrimental effects of these stresses [32]. For example, the expression of the gene *OsP5CS* is significantly upregulated in rice under high salt and drought conditions. In salt-tolerant rice varieties, the expression of *OsP5CS* is notably higher compared to salt-sensitive varieties, resulting in increased proline accumulation, which in turn enhances the plant’s stress tolerance [35]. Similarly, *OsTPS1* is another gene that is induced by various stressors, including drought, high salinity, low temperatures, and abscisic acid (ABA) treatments. In transgenic rice lines overexpressing *OsTPS1*, the concentrations of trehalose and proline are elevated compared to wild-type plants, and several stress-responsive genes such as *WSI18*, *RAB16C*, *HSP70*, and *ELIP* are upregulated. These findings suggest that *OsTPS1* may enhance abiotic stress tolerance in rice by not only increasing the levels of critical osmolytes like trehalose and proline but also by modulating the expression of key stress-responsive genes [36].

### 3.2. Regulation of Ion Balance

Although drought stress does not directly involve salt ions, plants regulate ion balance to adapt to both stresses. Under salt stress, plants need to exclude excess Na^+^ while maintaining K^+^/Na^+^ balance. High-affinity potassium transporters (HKT) have been shown to be crucial for maintaining K^+^/Na^+^ homeostasis in rice under salt stress [37]. The HKT gene family in rice includes *OsHKT1;5*, *OsHKT2;1*, and *OsHKT1;1*, which are located in the vascular bundles, parenchyma cells around the xylem vessels, cortical and endodermal cells of the roots, and phloem of the leaves, respectively [37,38,39]. These genes play essential roles in loading Na^+^ into the xylem, loading Na^+^ and K^+^ into the phloem, and the long-distance transport of Na^+^ and K^+^. The results demonstrated that under salt stress, *OsHKT1;5* actively transports excess Na^+^ from the shoots back to the roots through xylem unloading, thereby reducing Na^+^ toxicity and enhancing the salt tolerance of rice plants [37]. The expression of *OsHKT2;1* is upregulated in response to K^+^ starvation, but is rapidly downregulated when Na^+^ reaches toxic concentrations. Under conditions of K^+^ deficiency, *OsHKT2;1* specifically mediates the absorption of Na^+^ by the roots, utilizing it as a substitute nutrient to support plant growth [38]. Additionally, salt stress induces the expression of *OsHKT1;1*, which plays a critical role in minimizing Na^+^ accumulation in the shoots, thereby helping rice to better adapt to salt stress [39]. The transcription complex composed of *OsBAG4*, *OsMYB106*, and *OsSUVH7* regulates the expression of the key salt tolerance gene *OsHKT1;5* [37]. The *OsPRR73* protein binds to the promoter of *OsHKT2;1*, recruiting histone deacetylase HDAC10 to inhibit *OsHKT2;1* expression, thereby reducing Na^+^ uptake and avoiding excessive accumulation [40]. *OsMYBc* positively regulates salt tolerance in rice by binding to the AAANATNC(C/T) sequence in the *OsHKT1;1* promoter and upregulating its expression [41].

Studies on HKT genes in other crops have shown that HKT also positively affects drought tolerance. For example, in Arabidopsis *hkt1-1* mutants, overexpression of sorghum *SbHKT1;4*, *SbHKT1;5*, and *SbHKT2;1* genes significantly enhances drought tolerance in transgenic plants [42]. Maize *ZmHKT1* preferentially expresses in the root stele, altering the stele diameter to form thicker roots to cope with drought stress [43].

### 3.3. Regulation of the Antioxidant System

ROS are highly reactive by-products of normal cellular metabolism that readily react with lipids, proteins, and nucleic acids. Both salt and drought stress elevate ROS levels, leading to oxidative damage and severely affecting plant growth and development [44]. To prevent ROS accumulation, rice enhances the activity of the antioxidant system under these stresses, increasing the expression and activity of antioxidant enzymes (such as SOD, CAT, and APX) to remove ROS and protect cells from oxidative damage. The antioxidant system includes non-enzymatic components such as tocopherol, ascorbic acid, and glutathione, as well as enzymatic components including SOD, CAT, and enzymes of the ascorbate-glutathione cycle like APX, MDHAR, DHAR, and GSH [45].

Numerous genes have been reported to be involved in the antioxidant system to cope with stress. For example, *OsDjC46* enhances rice’s antioxidant defense capacity under drought and salt stress by regulating the expression and activity of SOD and CAT. Overexpression of *OsDjC46* increases tolerance to salinity and drought by enhancing SOD and CAT enzyme activities, while knockout lines are more sensitive to salt and drought stress with reduced SOD and CAT activities [46]. *OsCYBDOMG1* positively regulates rice’s salt tolerance, plant growth, and grain yield by affecting ascorbic acid biosynthesis and redox state [47]. *OsRLCK5* interacts with rice glutaredoxin *GRX20*, participating in the ascorbate-glutathione cycle to maintain protein stability and membrane integrity, enhancing rice’s resistance to high levels of ROS [48]. Through these complex physiological and molecular mechanisms, rice can maintain growth and productivity under salt and drought stress, demonstrating remarkable adaptability and stress tolerance.

## 4. Transcriptional Regulatory Networks

Recent genomic and transcriptomic studies have identified numerous key genes and regulatory networks associated with rice’s salt and drought tolerance [49,50]. Under salt and drought stress, some identical or similar TFs, such as members of the DREB, NAC, and bZIP families, are activated. These TFs enhance the plant’s stress adaptation by regulating the expression of downstream stress-responsive genes (Figure 4) [51].

### 4.1. DREB Transcription Factors

DREB TFs contain a conserved AP2/ERF DNA-binding domain of approximately 60–70 amino acids, which specifically binds to the DRE/CRT (Dehydration-Responsive Element/C-repeat) element with the core sequence (A/GCCGAC). DREB TFs help plants adapt to environmental stresses by regulating the expression of stress-responsive genes [52,53]. DREB TFs are divided into DREB1 and DREB2 subgroups, primarily involved in low-temperature response and drought, and high salt stress, respectively. Under salt and drought stress, DREB genes are induced. Transgenic plants overexpressing DREB genes exhibit enhanced stress tolerance by improving plant phenotypes, increasing the synthesis and accumulation of organic osmolytes, enhancing antioxidant enzyme activities, participating in oxidative stress responses, and regulating downstream stress genes [54]. For example, the AP2/EREBP TF gene *OsDREB1F* is induced by high salt, drought, cold stress, and ABA treatment, and transgenic rice and Arabidopsis show enhanced tolerance to high salt, drought, and low temperatures [55]. *OsDREB6* has transcriptional activation activity, specifically binds to the DRE cis-element, mediates the expression of specific genes, improves rice germination rates, affects the accumulation of osmolytes and ROS, and positively regulates rice tolerance to osmotic stress, salt stress, and cold stress [56].

### 4.2. NAC Transcription Factors

The NAC TF family is characterized by a conserved NAC domain usually located at the N-terminus of the protein, including the NAM domain (NAC TFs, ATAF1/2, CUC2) and a variable C-terminal transcriptional activation region. The NAC domain regulates target gene transcription through DNA binding [57]. *OsNAC45* plays a significant role under various abiotic stresses by regulating the expression of genes related to root POD activity and development. Overexpressing *OsNAC45* in transgenic rice enhances tolerance to salt and drought stress, potentially by reducing ABA’s inhibitory effect on root growth, mitigating stress inhibition of root growth, increasing lignin synthesis in roots, and reducing ROS accumulation [58]. *OsNAC2* regulates abiotic stress and ABA-mediated responses, acting as a connection point between ABA and abiotic stress pathways. Overexpression of *OsNAC2* decreases resistance under high salt and drought conditions, leading to yield reduction at the flowering stage [59].

### 4.3. bZIP Transcription Factors

Rice bZIP TFs are characterized by a conserved bZIP domain, consisting of a basic region and a leucine zipper region. The basic region contains the DNA binding site, while the leucine zipper region is used for TF dimerization [60]. *OsbZIP72* can directly bind to the promoters of *OsSWEET13* and *OsSWEET15* and activate their expression, regulating sucrose transport and distribution to respond to abiotic stress, helping maintain sugar homeostasis under drought and salt stress. *OsbZIP72* also binds to the ABA response element in the promoter region of the high-affinity potassium transporter gene *OsHKT1;1* and activates its expression, participating in ABA signaling pathways mediating salt and drought tolerance [61].

### 4.4. WRKY Transcription Factors

The WRKY TF family is a significant and diverse TF family in plants, playing key roles in regulating plant growth, development, disease defense, and environmental stress responses. WRKY TFs are named for their unique WRKY domain, which specifically binds to W-box elements (TTGAC[C/T]) [62]. *OsWRKY45* encodes a rice WRKY TF, with *OsWRKY45-1* and *OsWRKY45-2* being two alleles at this locus. These alleles have opposite functions in rice-Xoo interactions, mediating different defense response signaling pathways. Similarly, transgenic plants of these two alleles show differential expression in ABA and abiotic stress response genes but exhibit similar responses under cold and drought stress. *OsWRKY45-1* negatively regulates ABA signaling, while *OsWRKY45-2* positively regulates ABA signaling. Additionally, *OsWRKY45-2* positively regulates salt stress responses, while *OsWRKY45-1* negatively regulates them [63].

### 4.5. MYB Transcription Factors

MYB TFs have a DNA-binding domain composed of one to three repeat sequences (R1, R2, R3), each containing approximately 52 amino acids forming a helix-turn-helix structure. These repeats specifically bind to specific elements in DNA, such as MBS (MYB binding site). The C-terminus of MYB TFs typically contains transcriptional activation or repression domains for regulating downstream gene expression [64]. The MYB TF *OsMYB48* plays a critical role in rice’s drought and salt tolerance by regulating stress-induced ABA synthesis [65].

### 4.6. AP2/ERF Transcription Factors

The AP2/ERF family is named after APETALA2 (AP2) and Ethylene-Responsive Factor (ERF). Members of this family have one or more AP2/ERF DNA-binding domains. Based on the structure and number of these domains, they can be divided into four main subfamilies: AP2, ERF, RAV, and Soloist. Studies have found that ERF and RAV are mainly involved in stress responses [66]. Salt and PEG treatments can rapidly induce the expression of ERF and RAV TFs, regulating the expression of related stress-responsive genes and enhancing rice’s adaptability to stress. Through the regulation of these TFs, rice effectively responds to salt and drought stress, improving its survival and productivity [67].

## 5. Regulation by Plant Hormones

In addition to TFs, plant hormones are key factors influencing rice’s stress tolerance, playing a crucial role in adapting to various stresses (Figure 5A). Plant hormones are endogenous compounds that act as growth regulators in plants, functioning either at their site of synthesis or after translocation within the plant under different environmental and stress conditions [68].

### 5.1. Abscisic Acid (ABA)

Abscisic acid (ABA) is a vital plant hormone distributed widely in various plant tissues and organs, playing a crucial role in regulating plant growth and development. It is a key factor in plant responses to salt and drought stress, encompassing functions such as stomatal regulation, ion balance, stress-responsive gene expression, and metabolic changes [69]. Under stress conditions, the synthesis of abscisic acid (ABA) is significantly upregulated, which in turn initiates a critical signal transduction pathway. ABA binds to the PYR/PYL/RCAR receptors located on the cell membrane, triggering a cascade of molecular events. Upon binding, these receptors inhibit the activity of protein phosphatase 2C (PP2C), a key negative regulator in the pathway. This inhibition allows for the activation of SnRK2 kinase, which then phosphorylates the transcription factors ABF/AREB. Once phosphorylated, these transcription factors enter the nucleus and initiate the expression of stress-resistant genes, thereby enhancing the plant’s ability to cope with adverse conditions (Figure 5A) [70]. Many genes have been verified to mediate ABA synthesis and degradation in response to stress. *OsMLP423* is a positive regulator of drought and salt tolerance in rice, modulating abiotic stress tolerance through an ABA-dependent pathway. Transgenic rice overexpressing *OsMLP423* shows increased sensitivity to ABA and enhanced tolerance to drought and salt stress. Physiological analyses indicate that overexpressing *OsMLP423* may reduce membrane damage and ROS accumulation by regulating water loss rate and ABA-responsive gene expression under drought and salt stress [71]. Conversely, *OsCBE1* negatively regulates abiotic stress responses and the ABA signaling pathway, with mutants exhibiting significantly higher survival rates under salt, drought, and cold stress compared to wild-type and overexpression lines [72].

### 5.2. Gibberellins (GA)

Gibberellins (GA) are hormones critical for plant growth and also regulate growth under abiotic stress [73]. Generally, gibberellin (GA) binds to its receptor GID1 (Gibberellin Insensitive Dwarf1) to form the GA-GID1 complex. This complex interacts with the DELLA protein, promoting its degradation, which in turn regulates normal plant growth and development. However, under stress conditions, GA signaling can be inhibited, leading to the accumulation of DELLA proteins. This accumulation may interfere with the expression of genes involved in antioxidant responses, ion homeostasis, and cell protection. Some genes have been identified that enhance rice tolerance to stress by regulating GA metabolism and maintaining GA homeostasis under these challenging conditions (Figure 5B) [74]. *OsDSK2a* interacts with the GA deactivation enzyme EUI. Under salt stress, *OsDSK2a* levels decrease, leading to increased EUI accumulation, promoting GA metabolism, and reducing plant growth. Thus, *OsDSK2a* and *EUI* have opposing roles in regulating rice growth under salt stress through GA metabolism and homeostasis [75]. The GA catabolism pathway genes *OsGA2ox5* and *OsCYP71D8L* enhance plant salt tolerance by reducing GA accumulation through delayed growth, indicating a negative regulatory role of GA in rice salt tolerance [76].

### 5.3. Other Hormones

Other hormones, such as ethylene (ETH), cytokinins (CK), and jasmonic acid (JA), also play roles in environmental stress responses, participating in abiotic stress responses. Many genes have been verified to regulate these hormones and contribute to rice stress tolerance. *MHZ6* acts downstream of the ethylene signal positive regulator *OsEIN2* and positively regulates ethylene response in rice roots. Overexpression of *MHZ6* makes rice more sensitive to salt, indicating that *MHZ6* negatively regulates salt tolerance in rice [77]. *AGO2* regulates cytokinin distribution in plants by activating *BG3*, which enhances grain length and salt tolerance in rice. *BG3* encodes a purine permease involved in cytokinin transport. Overexpression of *AGO2* or *BG3* alters the spatial distribution of cytokinins, positively regulating grain length and ABA response, thus improving rice salt tolerance. Overexpression lines of *AGO2* show increased sensitivity to ABA [78]. *OsJAZ9* interacts with *OsCOI1a*, indicating its role in JA signaling regulation. As a transcriptional regulator, *OsJAZ9* interacts with *OsNINJA* and *OsbHLH* to form a transcriptional regulatory complex, modulating the expression of JA-responsive genes under salt stress. JA levels increase under salt stress. Upon JA perception, *SCFCOI1* recruits *OsJAZ9* for ubiquitination and degradation via the 26S proteasome, releasing *OsbHLH062* and *OsNINJA* to activate target gene expression. *OsbHLH062* activates downstream gene transcription, potentially further regulating adaptation to salt stress [79]. Through these complex regulatory mechanisms involving hormones, rice effectively responds to salt and drought stress, enhancing its adaptability and survival.

## 6. Breeding for Salt and Drought Tolerance in Rice

Breeding new rice varieties with enhanced tolerance to salt and drought stresses is a significant focus in current rice breeding research. Both salt and drought tolerance breeding primarily rely on conventional breeding techniques, which involve traditional artificial hybridization to introduce tolerance genes into elite rice varieties. Over multiple generations of stress screening and identification, breeders select lines with excellent comprehensive traits and stable inheritance (Table 1).

The first global efforts in screening and breeding salt-tolerant rice varieties began in 1939 when Sri Lanka developed the world’s first strongly salt-tolerant rice variety, Pokkali [88]. Subsequently, countries and institutions such as India and the International Rice Research Institute (IRRI) have conducted salt-tolerant germplasm screening, leading to the development and promotion of several salt-tolerant varieties. Chinese breeders have also employed conventional breeding methods and identified salt-tolerant germplasm resources. They have conducted salt-tolerant germplasm screening and variety selection under salt stress, resulting in the development and widespread agricultural application of rice varieties tolerant to varying salt concentrations [89].

Drought tolerance breeding follows a similar approach, involving the screening and identification of drought-tolerant germplasm as a basis for developing drought-tolerant varieties. Cultivated rice has two ecological types: upland rice and lowland rice. Upland rice, typically grown in high-altitude or drought-prone environments, has unique advantages in drought tolerance. Screening and systematic breeding of upland rice have led to the development of a series of drought-tolerant varieties [90].

In addition to conventional breeding, marker-assisted selection (MAS) is commonly used in developing stress-tolerant varieties. With the advancement of molecular marker technology and high-throughput sequencing, many stress-related quantitative trait loci (QTLs) have been identified, providing a foundation for fine-mapping, gene markers, and MAS in rice stress tolerance. Introducing major QTLs related to yield under stress conditions is a common strategy for improving stress tolerance in rice.

For example, researchers have used the salt-tolerant variety FL478 as a donor of the *Saltol* QTL for salt tolerance and the high-yielding variety PB1 as the recipient parent. Through two backcrosses and three generations of selfing, combined with foreground and background selection, they obtained 24 near-isogenic lines (NILs) with enhanced seedling-stage salt tolerance and similar traits to the parents [91]. Another study introduced the *Saltol* QTL from FL478 into the high-yielding but salt-sensitive variety ADT45, resulting in NILs that maintain high yields in both normal and saline fields [88]. Similarly, drought tolerance breeding involves the introduction of major drought QTLs. One study introduced the QTLs *qDTY2.1* and *qDTY3.1* into the recipient parent Pusa 44, resulting in 14 NILs with high background recovery rates and significantly better yield and grain quality under drought conditions compared to the parent [82]. Another study introduced the QTLs *qDTY2.2*, *qDTY3.1*, and *qDTY12.1* into MR219, resulting in NILs with higher yield and improved drought tolerance, with different QTL combinations showing varying performances [83].

Numerous studies have generated genetic populations through hybridization between tolerant and sensitive varieties, identifying many stress-related QTLs [91,92,93]. However, due to the complexity and polygenic regulation of stress tolerance in rice, many QTLs have not yet been applied in breeding, with most efforts focusing on improving existing high-yielding varieties. Further in-depth research on QTLs is needed in the future.

## 7. Future Research Directions and Application Prospects

Future rice research should integrate multi-omics technologies and genome editing techniques, combined with epigenetic regulation, microbial symbiosis, smart agriculture, and research on combined stresses, to develop multifunctional rice varieties that are tolerant to salt, drought, and heat. Additionally, breeding water-saving and drought-resistant rice (WDR) will enhance the efficiency of agricultural production in saline and drought-prone areas, providing new directions for sustainable agriculture (Figure 6).

### 7.1. Epigenetic Regulation

Research on DNA methylation, histone modifications, and small RNAs in stress responses can facilitate the development of epigenetic marker-assisted breeding techniques. Many studies have demonstrated that epigenetics play a crucial role in regulating important agronomic traits in crops. For example, the *OsSPL14* gene, which controls the ideal plant architecture in rice, is negatively regulated by *miR156*. A single nucleotide variation in the *miR156* target site weakens its regulatory effect, leading to moderate *OsSPL14* expression, ideal plant architecture, and high disease resistance [94]. Similarly, overexpression of *miR397* enhances its negative regulation of the laccase gene LAC, which is involved in lignin biosynthesis, significantly increasing spikelet number and seed size. Epigenetics also play a vital role in regulating stress tolerance; *miR168a-5p* targets *OsOFP3*, *OsNPF2.4*, and *OsAGO1a*, regulating seed length, nitrogen allocation, and salt tolerance, thereby providing a series of important agronomic traits for rice breeding [95].

### 7.2. Microbial Symbiosis

Microbes have shown great potential in enhancing rice salt tolerance. Many studies have indicated that certain rhizosphere microbes can alleviate Na^+^ ion toxicity, reduce ROS, and induce hormone regulation under salt stress, thus improving rice salt tolerance. Utilizing beneficial rhizosphere and endophytic microbes to enhance rice’s salt and drought tolerance is a promising method to increase yield in saline-alkali environments through microbiome research [96].

### 7.3. Smart Agriculture Technologies

With the development of intelligent technologies such as the Internet of Things (IoT), big data, cloud computing, and 5G, smart agriculture technologies are continuously evolving and integrating. Agricultural production is gradually becoming greener, standardized, digital, networked, and intelligent. The use of sensors, drones, and big data technologies to achieve precision irrigation and fertilization, optimizing field management, is a future direction to improve rice production efficiency in saline and drought-prone areas [97].

### 7.4. Research on Combined Stresses and Multi-Stage Stress Tolerance

Currently, most studies separately address salt and drought stresses, with few reports on the combined effects of salt and drought stress on rice growth and development and their physiological mechanisms. Additionally, most research focuses on short-term effects of stress, and systematic studies on the impacts at different growth stages (such as tillering, heading, and grain filling) throughout the entire growth period are lacking [98]. Investigating the comprehensive tolerance mechanisms of rice under multiple environmental stresses and developing multifunctional rice varieties with tolerance to salt, drought, and heat throughout the entire growth period is a crucial direction for future breeding [99].

Rice tolerance to salt and drought stress is a complex biological phenomenon that involves multiple levels of biological processes [100,101]. To effectively study these intricate processes, multi-omics approaches are essential, integrating fields such as genomics, transcriptomics, proteomics, metabolomics, and epigenomics [102,103]. By applying these multi-omics techniques in combination, researchers can achieve a comprehensive understanding of the biological response mechanisms in rice under salt and drought stress. For instance, a combined analysis of transcriptomics and metabolomics has revealed the molecular mechanisms by which alginate oligosaccharides (AOS) alleviate salt stress in rice seedlings [104]. Similarly, comparative profiling of transcriptomes and metabolomes has uncovered the complex molecular pathways underlying salt tolerance in rice seedlings, identifying key genes associated with this trait [105]. Moreover, through the integration of transcriptomic and metabolomic data, it has been demonstrated that OsCIPK17 positively regulates rice drought resistance by contributing to the accumulation of citric acid in the tricarboxylic acid cycle, offering new insights into enhancing drought tolerance in rice [106]. These findings not only advance fundamental research but also provide a solid scientific foundation for breeding rice varieties with improved tolerance to both salt and drought conditions.

### 7.5. Improvement of Water-Saving and Drought-Resistant Rice for Salt-Alkaline Tolerance

Water-saving and drought-resistant rice (WDR) is a new type of cultivated rice that combines the high-yield and quality traits of lowland rice with the water-saving and drought-resistant characteristics of upland rice [88]. Under irrigation, WDR yields and grain quality are comparable to lowland rice, but it can save more than 50% of water. In rainfed low-yield fields, WDR shows good drought resistance, allowing both water-saving cultivation in paddies and direct seeding in dry fields. Extensive promotion and application have demonstrated WDR’s high utilization value in dry fields. Using superior WDR germplasm as a foundation, breeding new germplasm with combined salt and drought tolerance through hybridization offers a new direction for agricultural production in saline and drought-prone areas [107].

## 8. Conclusions

Rice exhibits many similar physiological responses under salt and drought stress, including osmotic regulation, ion balance, antioxidant responses, signal transduction, and gene expression regulation. These similarities reflect the shared adaptive mechanisms and stress-resistance strategies plants employ when facing different environmental stresses. Many reported genes have been found to mediate physiological responses under both salt and drought stress, enhancing rice’s tolerance to these conditions and providing valuable references for breeding rice suitable for arid and saline soils.

In summary, research on rice salt and drought tolerance is of paramount importance for ensuring food security and sustainable agricultural development. In recent years, many key genes and regulatory networks associated with salt and drought tolerance have been identified, and the molecular mechanisms are gradually becoming clearer. However, further exploration of the functions of these genes and their complex regulatory networks is needed. Utilizing advanced genomic and molecular technologies, combined with ecological agricultural management practices, will aid in developing more efficient multi-stress-tolerant varieties and practical application strategies. This will provide a solid foundation for improving rice productivity and addressing global environmental changes.

## Figures and Tables

**Figure 1 ijms-25-09404-f001:**
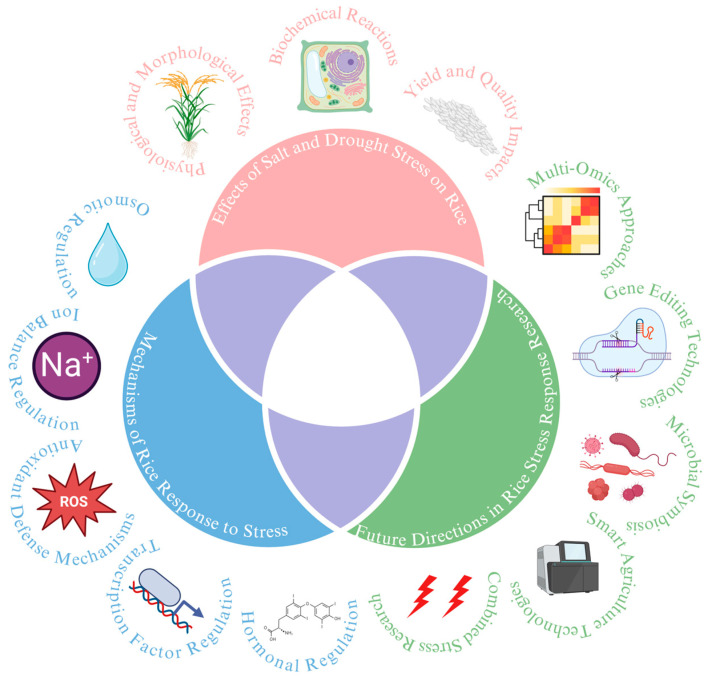
Comprehensive overview of rice responses to salt and drought stress: mechanisms, impacts, and future research directions (image created in BioRen-der.com accessed on 16 August 2024).

**Figure 2 ijms-25-09404-f002:**
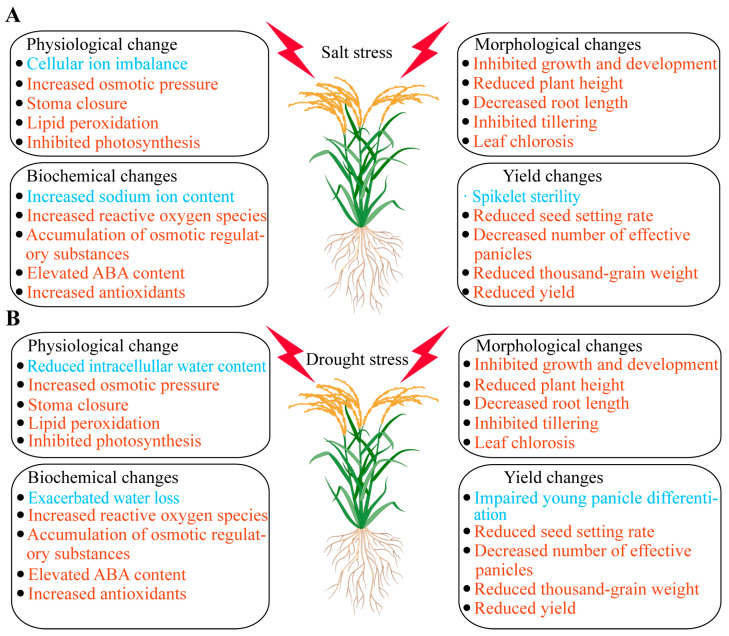
The image illustrates the physiological, biochemical, morphological, and yield changes in rice under salt stress (**A**) and drought stress (**B**). The red font indicates changes that occur under both types of stress, while the blue font highlights changes specific to each stress type.

**Figure 3 ijms-25-09404-f003:**
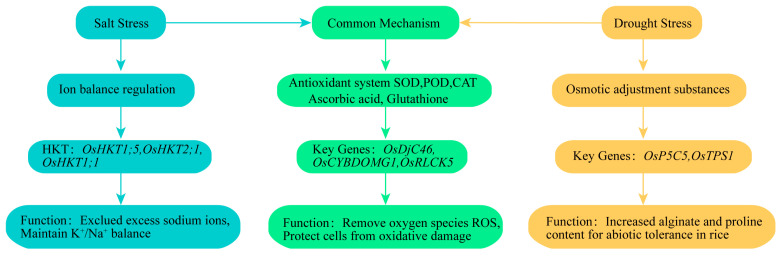
Mechanisms of rice responses to salt and drought stress. This diagram illustrates the mechanisms of rice responses to salt and drought stress, emphasizing both unique pathways and overlapping mechanisms. The color coding used is as follows: blue arrows and boxes represent pathways and mechanisms specific to salt stress, yellow arrows and boxes indicate those unique to drought stress, and green arrows and boxes highlight pathways shared by both salt and drought stress responses. This color distinction emphasizes the similarities and differences between the two types of stress and aids in understanding their interrelations.

**Figure 4 ijms-25-09404-f004:**
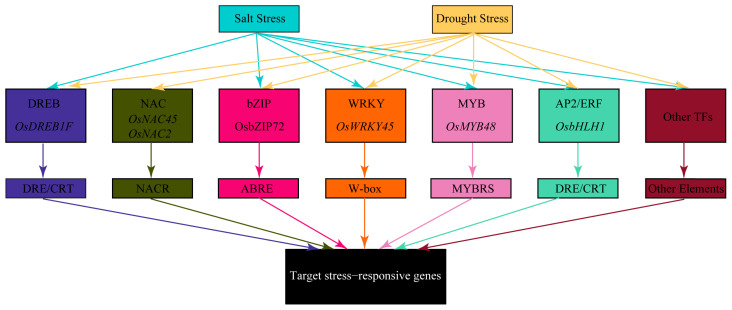
Transcription factor network in rice responding to salt and drought stress. This diagram illustrates the TF regulatory network in rice under salt and drought stress conditions. It includes several key TFs such as DREB, NAC, bZIP, WRKY, MYB, AP2/ERF, and other TFs, along with their specific binding elements like DRE/CRT, NACR, ABRE, W-box, MYBRS, and others. These TFs regulate the expression of target stress-responsive genes through these binding elements, aiding the plant in coping with environmental stress. The arrows represent the regulatory relationships between different TFs and their impact on target genes, highlighting the complex gene regulatory networks involved in plant stress resistance.

**Figure 5 ijms-25-09404-f005:**
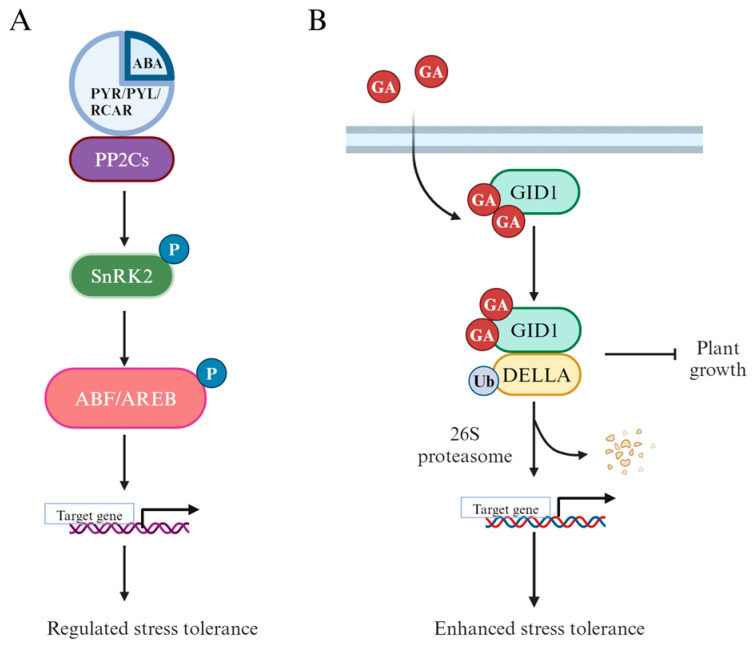
Schematic representation of the hormonal signaling pathways involved in stress tolerance and plant growth regulation in rice. (**A**) The ABA (abscisic acid) signaling pathway: ABA binds to its receptors (PYR/PYL/RCAR), which inhibits PP2C phosphatases, allowing the activation of SnRK2 kinases. Activated SnRK2 then phosphorylates and activates the ABF/AREB transcription factors, which regulate the expression of stress-responsive genes, enhancing stress tolerance. (**B**) The GA (gibberellin) signaling pathway: Gibberellin (GA) binds to its receptor GID1, leading to the degradation of DELLA proteins via the 26S proteasome pathway. The degradation of DELLA releases its inhibitory effect on plant growth, thereby promoting growth and also indirectly contributing to enhanced stress tolerance.

**Figure 6 ijms-25-09404-f006:**
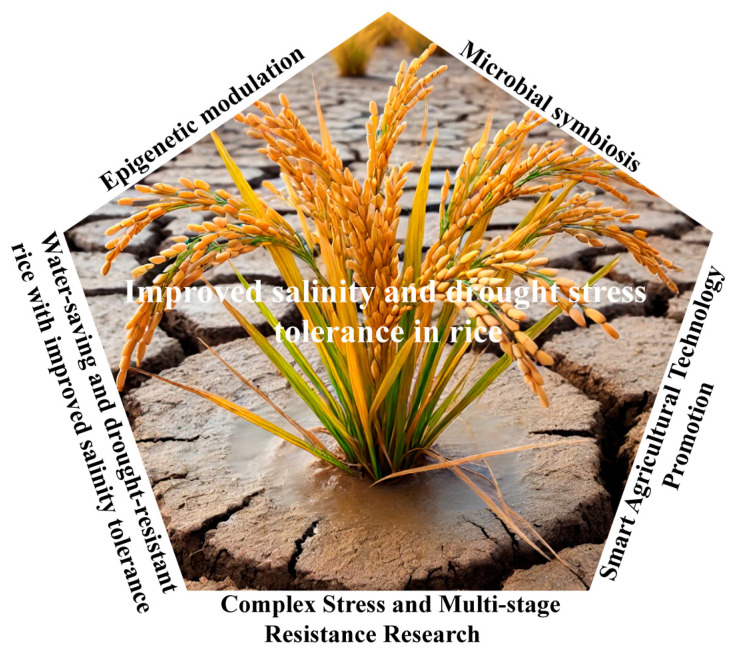
Improved salinity and drought stress tolerance in rice. The image depicts a robust rice plant thriving in cracked, dry soil, symbolizing resilience under salinity and drought stress conditions. Surrounding the central image are five key areas of research and development that contribute to enhancing rice tolerance to these stresses.

**Table 1 ijms-25-09404-t001:** Improved rice varieties with enhanced salt and drought tolerance.

QTL	Recipient Parent	Improvement Effect	Reference
*Saltol*	PusaBasmati1 (PB1)	24 NILs with enhanced seedling-stage salt tolerance and similar traits to the parent	[80]
*qDTY2.1*; *qDTY3.1*	Pusa 44	14 NILs with significantly better yield and grain quality under drought compared to the parent	[81]
*qDTY2.2*; *qDTY3.1*; *qDTY12.1*	MR219	NILs with higher yield and improved drought tolerance; different QTL combinations show varying performances	[82]
*qDTY3.1*; *qDTY6.1*; *qDTY6.2*	TDK1	High-yielding and drought-tolerant NILs	[83]
*qDTY1.1*; *qDTY2.1*; *qDTY3.1*; *qDTY11.1*	Samba Mahsuri	Higher yield under drought conditions	[84]
*qRL6.1*; *qRL12.1*	Hasawi × IR29	Increased root length under salt stress	[85]
*qPT3.1*	Hasawi × BRRI dhan28	Improved tillering under salt stress	[86]
*rkc3.1*; *rnc3.1*	Kalarata × Azucena	Enhanced root K^+^ and Na^+^ concentrations	[87]

## Data Availability

The original contributions presented in the study are included in the article; further inquiries can be directed to the corresponding authors.

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
