# Peer review of "Physiological and Molecular Mechanisms of Rice Tolerance to Salt and Drought Stress: Advances and Future Directions"

_ijms, 2024, doi:10.3390/ijms25179404_

Round 1

Reviewer 1 Report

Comments and Suggestions for Authors

In this review, the authors have focused on the physiological and molecular mechanisms associated with salt and drought stress tolerance in rice. They discuss the advances in molecular biology that reveal how rice copes with stress, including osmotic regulation, ion balance, antioxidant responses, signal transduction, gene expression regulation, transcription factors, and plant hormones. Additionally, the authors outline the CRISPR/Cas9 technology-mediated gene editing and suggest future research directions, such as integrating multi-omics approaches and utilizing smart agriculture technologies. This review paper will be useful to plant stress physiology researchers. However, the reviewer has some suggestions regarding this review. Thus, the authors need to consider the following comments to improve the quality of this manuscript.

The introduction section looks shallow. Improve it with more details. Provide a schematic representation of the review. Please refer the following paper for ease of understanding- www.mdpi.com/2079-7737/11/7/1022

Section 2.1 seems to literature needs to be improved. Please refer to new articles and cite the appropriate references for this section.

In CO2: 2 should be subscript. Check and revise it in the entire manuscript.

Follow the text reference formatting according to the journal.

Gene names should be in italics. Check and revise the same throughout the manuscript. Eg. Line 196: OsRLCK5

The reference to Figure 3 was not included in the text. Please verify and include the citation.

The authors should thoroughly check the entire manuscript's formatting, punctuation, spacing errors, and symbols. For example, on line 204, the authors have mentioned "transcription factors (TFs)" and line 218 is written in full form. They should abbreviate it and maintain consistency throughout the entire manuscript.

The focused review topic is broad. However, in the entire review paper, there are minimal literature references; please refer to more articles and cite them with enhanced details and information.

In section 5, adding a new figure stating the hormonal process/regulations related to focused stress mechanisms will be helpful to the readers.

If possible, authors can include salt and drought stress-related multi-omics approaches in the manuscript. 

Author Response

Comment 1: The introduction section looks shallow. Improve it with more details. Provide a schematic representation of the review. Please refer the following paper for ease of understanding- www.mdpi.com/2079-7737/11/7/1022.

Response: Thank you for your valuable suggestion. We carefully reviewed the recommended literature, particularly the paper from MDPI (www.mdpi.com/2079-7737/11/7/1022), which provided significant inspiration and insight. The depth and structure of that paper greatly informed our approach, leading us to revise the introduction with more comprehensive details. Specifically, we expanded the content in lines 25-65 to provide a deeper context and added a schematic representation to enhance the clarity and structure of the review. We have also cited this influential work in the revised manuscript, acknowledging its contribution to our understanding and the development of our review.

Comment 2: Section 2.1 seems to literature needs to be improved. Please refer to new articles and cite the appropriate references for this section.

Response: We appreciate your insightful feedback. In response, we have reviewed additional recent articles and have enriched Section 2.1 with more relevant references to provide a stronger foundation for the discussion. These additions aim to enhance the depth and scholarly value of the section.

Comment 3: In CO2: 2 should be subscript. Check and revise it in the entire manuscript.

Response: Thank you for bringing this to our attention. We have thoroughly checked the entire manuscript and have corrected all instances where subscripts were needed, including "CO2". This ensures consistency and accuracy throughout the document.

Comment 4: Follow the text reference formatting according to the journal.

Response: We sincerely appreciate your reminder. We have meticulously reviewed all references and have reformatted them to align with the journal's guidelines, ensuring that our submission meets the required standards.

Comment 5: Gene names should be in italics. Check and revise the same throughout the manuscript. Eg. Line 196: OsRLCK5

Response: Thank you for your careful review and valuable suggestion. We have gone through the entire manuscript and corrected the formatting of all gene names, including "OsRLCK5" on line 196, to ensure they are presented in italics as required.

Comment 6: The reference to Figure 3 was not included in the text. Please verify and include the citation.

Response: We appreciate your careful observation. We have verified and included the correct citation for Figure 4, now referred to as "Transcription Factor Network in Rice Responding to Salt and Drought Stress," in line 253 of the revised manuscript.

Comment 7: The authors should thoroughly check the entire manuscript's formatting, punctuation, spacing errors, and symbols. For example, on line 204, the authors have mentioned "transcription factors (TFs)" and line 218 is written in full form. They should abbreviate it and maintain consistency throughout the entire manuscript.

Response: Thank you for your detailed feedback. We have conducted a thorough review of the manuscript to correct all formatting, punctuation, and spacing errors. Additionally, we ensured consistency in the use of abbreviations and symbols throughout, including the consistent use of "transcription factors (TFs)" where appropriate.

Comment 8: The focused review topic is broad. However, in the entire review paper, there are minimal literature references; please refer to more articles and cite them with enhanced details and information.

Response: We are grateful for your suggestion. In response, we have significantly expanded the literature review, adding numerous relevant references and enriching the discussion with enhanced details and information. These changes are reflected in lines 32-54, 72-126, and 513-528 of the revised manuscript.

Comment 9: In section 5, adding a new figure stating the hormonal process/regulations related to focused stress mechanisms will be helpful to the readers.

Response: Thank you for your excellent suggestion. We have incorporated a new figure in Section 5 that outlines the hormonal processes and regulations related to the stress mechanisms discussed. This addition aims to provide readers with a clearer visual understanding of the complex interactions involved.

Comment 10: If possible, authors can include salt and drought stress-related multi-omics approaches in the manuscript.

Response: We greatly appreciate your insightful recommendation. In Section 7.4 (lines513-528), we have included a discussion on the application of multi-omics approaches to understanding saline and drought stress in rice. This addition provides a more comprehensive view of current research methodologies in this area.

Reviewer 2 Report

Comments and Suggestions for Authors

Comment 1: Line 50-53, the sentence discussing the dual challenges of salt and drought stress is lengthy and somewhat complex. It would be clearer if you could split it into two shorter, more focused sentences.

Comment 2: Please add a citation for the statement on line 89 regarding the increased levels of antioxidant enzymes (such as superoxide dismutase, peroxidase, and catalase) in rice under salt stress.

Comment 3: Line 134, please provide a reference or additional details on the role of OsP5CS and OsTPS1 in enhancing rice stress tolerance, particularly regarding their specific impact under high salt and drought conditions.

Comment 4: Line 160, consider discussing any key studies or protocols that describe the use of high-affinity potassium transporters (HKT) in maintaining K+/Na+ homeostasis in rice under salt stress. This will enhance the reader’s understanding of the significance of these genes.

Comment 5: The description of the impact of salt and drought stress on rice morphology and physiology (lines 175-200) could benefit from a brief comparison to other crops affected by similar stresses. This would provide broader context and reinforce the relevance of the findings.

Comment 6: The diagrams in the manuscript, particularly Figure 2 (Mechanisms of Rice Responses to Salt and Drought Stress), could be more informative with additional labeling or a brief description of the color coding and arrows used. This will help readers better interpret the information.

Comment 7: Line 220, the manuscript briefly mentions the potential of CRISPR/Cas9 technology in enhancing rice stress tolerance. It would be valuable to include examples of successful applications or ongoing research in this area to illustrate the potential of this technology more concretely.

Author Response

Comment 1: Line 50-53, the sentence discussing the dual challenges of salt and drought stress is lengthy and somewhat complex. It would be clearer if you could split it into two shorter, more focused sentences.

Response: Thank you for your valuable suggestion. We have revised the text by splitting the original sentence into two more focused sections, with one addressing the morphological impacts and the other describing the physiological and biochemical responses. These changes are reflected in lines 127-137, making the content clearer and easier to understand.

Comment 2: Please add a citation for the statement on line 89 regarding the increased levels of antioxidant enzymes (such as superoxide dismutase, peroxidase, and catalase) in rice under salt stress.

Response: We appreciate your suggestion. We have now included the appropriate literature citation that discusses the increased levels of antioxidant enzymes like superoxide dismutase, peroxidase, and catalase in rice under salt stress. This addition can be found in lines 98-101 of the revised manuscript.

Comment 3: Line 134, please provide a reference or additional details on the role of OsP5CS and OsTPS1 in enhancing rice stress tolerance, particularly regarding their specific impact under high salt and drought conditions.

Response: Thank you for pointing this out. We have enriched the manuscript with a more detailed description of the roles of OsP5CS and OsTPS1 in enhancing rice stress tolerance, particularly under high salt and drought conditions. This information, along with the appropriate references, is now included in lines 186-195.

Comment 4: Line 160, consider discussing any key studies or protocols that describe the use of high-affinity potassium transporters (HKT) in maintaining K+/Na+ homeostasis in rice under salt stress. This will enhance the reader’s understanding of the significance of these genes.

Response: Thank you for your valuable suggestion. We have expanded the manuscript by adding more detailed discussions and references related to high-affinity potassium transporters (HKT) and their role in maintaining K+/Na+ homeostasis in rice under salt stress. These additions can be found in lines 205-212, and they aim to provide a deeper understanding of the significance of these genes.

Comment 5: The description of the impact of salt and drought stress on rice morphology and physiology (lines 175-200) could benefit from a brief comparison to other crops affected by similar stresses. This would provide broader context and reinforce the relevance of the findings.

Response: Thank you for your insightful suggestion. In this review, our focus is primarily on summarizing the specific changes in rice under salt and drought stress, with the goal of aiding rice breeding for enhanced tolerance. While we acknowledge the broader relevance, discussing other crops falls outside the scope of our current review. However, we believe our concentrated focus on rice will still provide valuable insights.

Comment 6: The diagrams in the manuscript, particularly Figure 2 (Mechanisms of Rice Responses to Salt and Drought Stress), could be more informative with additional labeling or a brief description of the color coding and arrows used. This will help readers better interpret the information.

Response: Thank you for your valuable comments. We have taken your suggestion to heart and have updated Figure 3 by adjusting the colors of the arrows to better correspond to the different stress pathways. We have also added new labels to the diagram, along with a brief description of the color coding and arrows used. These changes will help readers more easily interpret the information presented.

Comment 7: Line 220, the manuscript briefly mentions the potential of CRISPR/Cas9 technology in enhancing rice stress tolerance. It would be valuable to include examples of successful applications or ongoing research in this area to illustrate the potential of this technology more concretely.

Response: Thank you for your valuable suggestion. CRISPR/Cas9 technology is indeed a powerful tool for constructing mutants of target genes and performing functional analyses. In response to your comment, we have included additional examples and discussions of successful applications and ongoing research involving CRISPR/Cas9 in enhancing rice stress tolerance. These updates are reflected in lines 219-223, 290-293, and 358-361, providing a more concrete illustration of the potential of this technology.

Round 2

Reviewer 1 Report

Comments and Suggestions for Authors

The authors have suitably incorporated my suggestions in the revised manuscript. Now, the manuscript quality has improved more than before. However, the manuscript still needs some minor changes before acceptance. Therefore, the authors need to consider the following comments to improve the quality of this manuscript.

Line 36: The scientific name of the plant should be in italics. Check the entire manuscript and revise it.

The authors should thoroughly check the entire manuscript’s formatting, punctuation, spacing errors, and symbols. For example, line 43 has no space between the reference and the next sentence.

Line 65: It seems Figure 1. Figure 1. Please correct it.

Line 65, 171, 252, 466: Why is the caption of figures 1, 3, 4, and 6, each word is capitalized? It is not needed. Check and revise it.

Author Response

Comment 1: Line 36: The scientific name of the plant should be in italics. Check the entire manuscript and revise it.

Response: Thank you for pointing out this formatting issue. We have thoroughly reviewed the entire manuscript and ensured that all scientific names, including the one on line 36, are now correctly formatted in italics.

Comment 2: The authors should thoroughly check the entire manuscript’s formatting, punctuation, spacing errors, and symbols. For example, line 43 has no space between the reference and the next sentence.

Response: We appreciate your attention to detail. We have conducted a comprehensive review of the manuscript to correct any formatting, punctuation, and spacing errors. The issue on line 43, where there was no space between the reference and the following sentence, has been corrected, along with any other similar issues throughout the manuscript.

Comment 3: Line 65: It seems Figure 1. Figure 1. Please correct it.

Response: Thank you for bringing this to our attention. The duplication error in line 65, where "Figure 1" was repeated, has been corrected. We have carefully reviewed all figure references in the manuscript to ensure there are no similar mistakes.

Comment 4: Line 65, 171, 252, 466: Why is the caption of figures 1, 3, 4, and 6, each word is capitalized? It is not needed. Check and revise it.

Response: We apologize for the inconsistency in figure captions. The capitalization of each word in the captions for Figures 1, 3, 4, and 6 has been revised to standard sentence case format, as per the journal's guidelines. We have also checked and ensured consistency across all figure captions in the manuscript.

Reviewer 2 Report

Comments and Suggestions for Authors

The present paper can be accepted.

Author Response

Thank you very much for your positive feedback and for recognizing the value of our work. We greatly appreciate your thorough review and are delighted that you find the manuscript suitable for publication. Your constructive comments have significantly contributed to improving the quality of our paper.